# The role of implementation climate in shaping early essential newborn care practice: Insights from a multi-center cross-sectional study in China

Hongxiao He[1], Jiahe Li[1], Junying Li[2], Hong Lu[1], Jie Lu[3], Linlin Cao[4], Luxia Gong[5], Ruyan Pang[5], Xiu Zhu[1]*

1 School of Nursing, Peking University, Beijing, China, 2 Handan Vocational College of Science and Technology, Hebei, China, 3 Beijing Obstetrics and Gynecology Hospital, Capital Medical University (Beijing Maternal and Child Health Care Hospital), Beijing, China, 4 Department of Gynaecology and Obstetrics, Peking University Third Hospital, Beijing, China, 5 Chinese Maternal and Child Health Association, Beijing, China

* zhuxiu@bjmu.edu.cn

## Abstract

### Background

The World Health Organization (WHO) recommends Early Essential Newborn Care (EENC) to improve newborn outcomes. However, uptake remains suboptimal in many low-resource settings. Organisational factors, such as implementation climate, are crucial but understudied in relation to EENC implementation.

### Objective

To explore how implementation climate mediates the relationship between knowledge, attitudes, and EENC practices.

### Design

Multi-site, cross-sectional study.

### Setting

Twelve tertiary maternity hospitals in China (December 2022–April 2023).

### Participants

433 nurse-midwives.

**Data availability statement:** All relevant data are within the paper and its Supporting information files.

**Funding:** This study was fund by a joint project between United Nations International Children's Emergency Fund and the China Maternal and Child Health Association (No: CMCHA/XM.2021/048). XZ received the award. The funders were not involved in the study design, data collection and analysis, decision to publish, or preparation of the manuscript.

**Competing interests:** The authors have declared that no competing interests exist.

## Methods

Validated questionnaires were use to assess knowledge, attitudes, practices, and perceived implementation climate related to EENC. Path analysis and logistic regression were employed to explore direct and indirect relationships.

## Results

A total of 69.3% participants reported good EENC practice. Significant predictors included good knowledge (adjusted odds ratio [AOR] = 2.75; 95% confidence interval [CI]: 1.76–4.31), positive attitudes (AOR = 2.00; 95% CI: 1.17–3.41), in-service training (AOR = 1.88; 95% CI: 1.17–3.02), holding a middle leadership role (AOR = 2.24; 95% CI: 1.20–4.17), and perceived workload. Nurse-midwives who reported heavier workloads were 48% less likely to hold positive attitudes towards EENC (AOR = 0.52; 95% CI: 0.28–0.94), which subsequently affected their EENC practice. The mean score of implementation climate was moderately favorable (3.30 ± 0.77), with the lowest in the *rewards* domain (3.02 ± 1.11). A one-point increase in climate score was associated with significantly higher odds of a positive attitude (AOR = 4.56; 95% CI: 2.98–6.99). Implementation climate influenced EENC practice indirectly through attitudes (RMSEA = 0.039).

## Conclusions

This study highlights the importance of both individual factors and organizational climate in shaping EENC practices. To improve EENC implementation, healthcare systems should prioritize enhancing the implementation climate through leadership support, establishing appropriate reward systems, and addressing workload challenges. Additionally, integrating EENC training into continuous professional development programs and strengthening support for mid-level leadership are key strategies.

## 1. Introduction

To reduce preventable stillbirths and neonatal deaths by 2035, the World Health Organization (WHO) and UNICEF (United Nations International Children's Emergency Fund) launched the *Action Plan for Healthy Newborn Infants in the Western Pacific Region (2014–2020)* [1]. A key focus is Early Essential Newborn Care (EENC)—a package of evidence-based interventions that includes immediate drying, delayed cord clamping, skin-to-skin contact, and kangaroo mother care for preterm infants. Evidence shows that EENC can reduce neonatal deaths by 22%−33%, decrease rates of postpartum haemorrhage, and improve breastfeeding outcomes [2,3].

Despite these benefits, EENC implementation remains limited. By 2022, only 4 of 9 countries in the Western Pacific had reached WHO's target of 80% EENC coverage. In China, where WHO and UNICEF have supported EENC scale-up since 2016, gaps persist in key practices such as delayed cord clamping and kangaroo mother care [4,5].

Previous studies have identified common barriers, including limited training, knowledge gaps, and heavy workloads [6–8]. However, most of this research has focused on individual-level factors. Less attention has been paid to organizational factors—such as implementation climate, which refers to staff perceptions of institutional support, expectations, and rewards for using evidence-based practices. A strong implementation climate is believed to improve practice by aligning staff values with innovations and fostering positive attitudes [9]. According to the foundational work by Klein and Sorra, climate efficacy hinges on "innovation-values fit" among staff [10], which is expected to promote favorable attitudes, thereby enhancing implementation behavior and improved outcomes [11].

Although qualitative work from Sri Lanka suggests that incentive structures may help sustain EENC [12], quantitative studies on the role of implementation climate in maternal-newborn care are scare. This gap persists despite the availability of validated implementation climate measures in behavioral health research [11,13] and theoretical frameworks linking climate to clinician attitudes and guideline adherence [14,15]. Previous research has strengthened the notion that implementers' intrinsic positive motivation collectively represented a strong, overall determinant of the uptake of guidelines to improve obstetric care [16].

Prior quantitative studies have relied heavily on linear regression models [17,18], which are unable to disentangle the interdependencies among organizational context, individual competencies (e.g., knowledge, attitudes), and practical actions. Path analysis allows researchers to examine both direct and indirect effects within these systems. Although increasingly used in mental health implementation studies [15], this method has rarely been applied to maternal and newborn care.

This study explores whether implementation climate influences EENC practice through its effect on providers' attitudes. Findings may help inform strategies to bridge the gap between EENC guidelines and real-world practice in maternity settings.

## 2. Materials and methods

A cross-sectional, multi-centre survey was conducted between December 2022 and April 2023 among nurse-midwives working in twelve tertiary-level maternity facilities across China. The study protocol received ethical approval from the Peking University Institutional Review Board (IRB00001052-22047) on 30 May 2022.

### 2.1. Study settings

The study was conducted across twelve tertiary-level maternity facilities, which are in-service midwifery training bases accredited by the Chinese Maternal and Child Health Association (CMCHA) [19]. These facilities included general hospitals, maternal and child health hospitals, and specialized maternity hospitals. They were purposively selected to ensure geographic and socioeconomic diversity.

**Geographic distribution:** Facilities were located in both eastern provinces (Beijing, Tianjin, Shanghai, Zhejiang, Jiangsu, Guangdong) and western provinces (Sichuan, Shanxi), reflecting diverse regional healthcare environments.

**Institutional characteristics:** All participating facilities were CMCHA-accredited training centres with annual delivery volumes ranging from 10,000–30,000 births, highlighting their roles as high-volume, regionally influential institutions.

**Socioeconomic context:** The selected facilities covered both economically developed and underdeveloped regions, ensuring representation of varying levels of healthcare resources. This sampling framework aimed to capture diversity in clinical practices, institutional capacity, and regional health priorities, thereby enhancing the generalizability of findings to tertiary maternity settings in China.

### 2.2. Sampling

The study employed a cluster sampling design with 12 healthcare facilities as discrete sampling units. The sample size was calculated using the single population proportion formula, adopting skin-to-skin contact implementation rate (83.6%) as the proxy indicator for EENC practice based on historical data [20]. Calculations assumed 95% confidence level with

5% margin of error (d = 0.05), yielding a minimum requirement of 207 participants. This baseline was increased by 10% to 228 to compensate for anticipated non-response.

Eligibility criteria required participants to: 1) hold valid national Maternal and Child Health Care Technology Certification for autonomous practice, 2) possess ≥one year of clinical experience in labor/delivery units, and 3) provide informed consent. Exclusion criteria were: interns, temporary trainees, and staff with extended leave (>three months) to ensure clinical competency and decision-making capacity.

All qualified nurse-midwives across facilities received participation invitations, resulting in 433 valid responses from 512 eligible practitioners (84.6% response rate). Primary non-response justification involved clinical shift time constraints. The achieved sample size not only surpassed initial requirements but also satisfied statistical power thresholds for advanced analyses including path modeling and logistic regression, ensuring robust detection of both direct effects and mediated relationships.

## 2.3. Measurement tool

A structured questionnaire was developed based on WHO guidelines [21] and previous studies [17,18], and was pilot-tested before the formal survey to ensure clarity and relevance. The questionnaire collected information on socio-demographic characteristics (including age, gender, fertility status, professional position, education level, years of experience in labour rooms, perceived workload, routine workflow, and prior EENC training), as well as constructs related to implementation climate, knowledge, attitudes, and practices regarding EENC. The complete questionnaire is available in S1 File.

The Implementation Climate Scale (ICS), developed by Ehrhart et al. [22], was used to assess implementation climate. This scale comprises 18 items across six dimensions: *focus*, *educational support*, *recognition*, *rewards*, *selection for evidence-based practice (EBP)*, and *selection for openness*. The Chinese version of the ICS, which has been validated among Chinese nurses [23], was applied in this study. The scale demonstrated excellent internal consistency, with a Cronbach's alpha coefficient of 0.971.

Knowledge of EENC was evaluated using nine items derived from WHO guidelines and a validated tool employed in Ethiopia [18,21]. Responses were recorded on a three-point scale: "agree", "not sure", and "disagree". Each response aligned with WHO recommendations was assigned a score of 1, while incorrect or uncertain responses received a score of 0. Total scores ranged from 0 to 9. Consistent with the approach used by Ashenef et al. [18], the mean score served as the cut-off: scores above the mean indicated good knowledge, while scores at or below the mean reflected poor knowledge.

Attitudes toward EENC were assessed using ten self-developed items. The first five items targeted attitudes toward EENC interventions, and the remaining five assessed attitudes toward general EENC implementation. Responses were rated on a 5-point Likert scale (0 = completely disagree to 4 = completely agree), with higher scores reflecting more positive attitudes. The Cronbach's alpha for this scale was 0.825. The tool was designed based on a combination of statistical analysis, qualitative insights, and expert review. Two items (items 4 and 5) were specifically included to capture attitudes toward controversial EENC interventions—non-routine disinfection of the cord stump and non-traditional suctioning of the mouth and nose—as these were areas of concern raised by local midwives during preliminary qualitative work. Item 10 was retained to reflect perceived competitive pressure and institutional motivation, an aspect of contextual attitude not fully covered by other items. Except for items 4, 5, and 10, which had modest factor loadings, all other items exhibited standard regression weights ranging from 0.596 to 0.905, exceeding the threshold of 0.5, thus supporting the scale's validity. Drawing on the Chinese version of the Modified Barthel Index and Bloom's criteria [24,25], participants scoring more than 80% (32/40) were considered to have a positive attitude; those scoring ≤80% were considered to have a negative attitude [26,27].

Practice of EENC was measured using ten items adapted from WHO recommendations [21] and Tewodros et al. [17]. Each item used a 4-point scale: never, seldom, sometimes, and always. Among them, "never" and "seldom" were recoded

as "not performed"; "sometimes" and "always" as "performed". Responses consistent with WHO guidelines were scored as 1; otherwise, as 0. Practice scores ranged from 0 to 10. Scores ≥8 indicated good practice; scores <8 indicated poor practice.

## 2.4. Data collection

Data were collected using self-administered online questionnaires distributed via the Wenjuanxing platform. Prior to participation, all respondents reviewed detailed study information and provided electronic informed consent before accessing the questionnaire. The survey link was distributed to eligible participants via the head nurses of labor and delivery units at the twelve participating facilities, who coordinated the dissemination of the electronic questionnaire within their teams. Throughout the data collection period, response progress and data quality were monitored in real time via the platform's backend. All responses were securely stored on Wenjuanxing's encrypted server incompliance with national data protection standards. Access to the data was limited to authorized research team members to ensure confidentiality and data security. No personally identifiable information was collected at any stage of the study.

## 2.5. Data management and statistical analysis

Data were cleaned and validated prior to analysis. Incomplete or invalid responses and outliers were removed to ensure data quality and accuracy. All statistical analyses were conducted using IBM SPSS Statistics for Windows, Version 26.0 (Armonk, NY, USA).

Descriptive statistics (frequencies and percentages) were used to summarize participants' background characteristics. Adjusted odds ratios (AORs) and 95% confidence intervals (CIs) were reported. Collinearity among independent variables was assessed, and variance inflation factor (VIF) values were all < 10, indicating no significant multicollinearity [28]. Univariable analysis was first conducted, and variables with a p-value <0.20 were included in the multivariable model [29]. The final model was selected using the likelihood ratio (LR) backward stepwise method. Model fit was assessed using the Hosmer-Lemeshow goodness-of-fit test. Statistical significance was set at a two-sided p-value <0.05.

Path analysis was conducted using AMOS version 24.0 (IBM Corp.) to explore the interrelationships among variables. Model fit was evaluated using the Goodness-of-Fit Index (GFI), Adjusted Goodness-of-Fit Index (AGFI), Normed Fit Index (NFI), Comparative Fit Index (CFI), Tucker-Lewis Index (TLI), and Root Mean Square Error of Approximation (RMSEA). GFI, AGFI, NFI, CFI, and TLI values >0.90 and >0.95 were considered indicative of acceptable and good model fit, respectively. RMSEA values <0.08 and <0.05 reflected marginal and good model fit, respectively. Squared multiple c orrelations ($R^2$) were used to assess the explanatory power of endogenous variables. To assess indirect effects, a bias-corrected bootstrap method with 1,000 resamples and 95% CIs was applied. All statistical tests were two-tailed, with p-values <0.05 considered statistically significant.

## 3. Results

### 3.1. Characteristics of the nurse-midwives

A total of 433 nurse-midwives participated in the survey. The majority of participants were female (424, 97.9%), with 9 (2.1%) males. The mean age of the participants was 34.3 ± 18.5 years. Most of the participants held a bachelor's degree (394, 91%), 27 (6.2%) held a specialty degree, and 12 (2.8%) held a master's degree. Among the participants, 13 (3%) held the position of nurse supervisor and 93 (21.5%) were group leaders, while the majority (327, 75.5%) did not hold any managerial role. Over half of the participants (260, 56%) had received training in EENC. In terms of experience in the labor room, 143 participants (33%) had worked for fewer than seven years, 196 (45.3%) had 8–14 years of experience, and 94 (21.7%) had more than 15 years of experience. The distribution of these characteristics is shown in Table 1.

**Table 1. Characteristics of the participants (n = 433).**

| Variables | Categories | Frequency | Percent (%) |
|---|---|---|---|
| Age (years) | | | |
| | < 25 | 18 | 4.2 |
| | 25-29 | 106 | 24.5 |
| | 30-34 | 163 | 37.6 |
| | ≥ 35 | 146 | 33.7 |
| Gender | | | |
| | Female | 424 | 97.9 |
| | Male | 9 | 2.1 |
| Parental status | | | |
| | Without child | 162 | 37.4 |
| | With child | 271 | 62.6 |
| Level of education | | | |
| | Collage | 27 | 6.2 |
| | Bachelor | 394 | 91.0 |
| | Master | 12 | 2.8 |
| Position | | | |
| | None | 327 | 75.5 |
| | Group leader | 93 | 21.5 |
| | Nurse supervisor | 13 | 3.0 |
| Years of working (years) | | | |
| | ≤ 7 | 143 | 33.0 |
| | 8~14 | 196 | 45.3 |
| | ≥ 15 | 94 | 21.7 |
| Routine workflow on EENC | | | |
| | Yes | 384 | 88.7 |
| | No | 49 | 11.3 |
| Perceived workload | | | |
| | Acceptable | 290 | 67.0 |
| | Heavy | 143 | 33.0 |
| Training of EENC | | | |
| | No | 117 | 27.0 |
| | Yes | 316 | 73.0 |

### 3.2. Nurse-midwives' knowledge, attitude and practice of EENC

Table 2 presents the knowledge, attitude, and practice regarding EENC among the nurse-midwives. Detailed responses to each question are provided in S2 File.

The mean knowledge score was 6.91 ± 1.15 ranging from 3 to 9. Among the participants, 293 (67.7%) demonstrated good knowledge of EENC. However, 36.3% were unaware that an inability to suckle or cry could be a red flag for a new-born. Additionally, nearly half (47.3%) agreed with the routine disinfection of the umbilical cord stump, and most (71.1%) disagreed with the practice of routine eye care to prevent infection.

The mean attitude score towards EENC was 35.58 ± 4.81 ranging from 14 to 40, with 355 (82.0%) participants exhibiting a positive attitude towards EENC. The item with the lowest mean score pertained to the safety of not disinfecting the umbilical cord stump (2.97 ± 1.23).

**Table 2. Knowledge, attitude, and practice of EENC among nurse-midwives (n = 433).**

|  | Frequency (%) | Mean (SD) |
|---|---|---|
| Knowledge of EENC |  | 6.91 (± 1.15) |
| Poor knowledge | 140 (32.3) |  |
| Good knowledge | 293 (67.7) |  |
| Attitude of EENC |  | 35.58 (± 4.81) |
| Negative | 78 (18.0) |  |
| Positive | 355 (82.0) |  |
| Practice of EENC |  | 7.95 (± 1.55) |
| Poor practice | 133 (30.7) |  |
| Good practice | 300 (69.3) |  |

Regarding practice, 300 (69.3%) participants were assessed as having good practice. The overall mean practice score was 7.95 ± 1.55 ranging from 2 to 10. A total of 334 (77.1%) participants did not routinely suction the newborn's nose or mouth and 383 (88.5%) ensured skin-to-skin contact immediately after birth. However, more than 55.2% continued to disinfect the umbilical cord stump. The majority (86.4%, 374 participants) maintained skin-to-skin contact for 90 minutes post-delivery. Baby-mother contact was not interrupted by routine care such as physical examinations or body measurements, according to 348 (80.4%) participants. Nevertheless, less than half (43.2%, 187 participants) performed routine eye care, such as administering erythromycin, to prevent infection.

### 3.3. Implementation climate score

The overall mean score for the implementation climate scale was 3.30 ± 0.77. The scores for the six dimensions of implementation climate were as follows: *focus* (3.50 ± 0.70), *educational support* (3.40 ± 0.80), *recognition* (3.25 ± 0.90), *rewards* (3.02 ± 1.11), *selection* (3.27 ± 0.92), and *selection for openness* (3.36 ± 0.84). Among these dimensions, *rewards* received the lowest score.

### 3.4. Factors associated with the attitude and practice of EENC

Logistic regression analysis identified several factors associated with nurse-midwives' attitudes toward EENC, including implementation climate, receipt of EENC training, knowledge of EENC, and workload. Per one-point increase in the implementation climate scale score, the odds of having a positive attitude increased by 4.56 times (AOR = 4.56, 95%CI: 2.98–6.99). Nurse-midwives who had received EENC training were 2.47 times (AOR = 2.47, 95%CI: 1.31–4.67) more likely to have a positive attitude toward EENC. Compared to those with poor knowledge, those with good knowledge were 2.23 times more likely to have a positive attitude (AOR = 2.23, 95%CI: 1.20–4.13). Heavier workloads were associated with a negative attitude, as those with heavier workloads were 48% less likely to hold a positive attitude (AOR = 0.52, 95%CI: 0.28–0.94). The univariable and binary logistic regression analyses for EENC attitude are shown in Table 3.

For EENC practice, logistic regression analysis revealed that knowledge of EENC, position, attitude toward EENC, and EENC training were significant associated factors. Participants with good knowledge of EENC were 2.75 times more likely to report good EENC practices than those with poor knowledge (AOR = 2.75, 95%CI: 1.76–4.31). Group leaders were 2.24 times more likely to report good practices compared to those without managerial positions (AOR = 2.24, 95%CI: 1.20–4.17). Those with a positive attitude toward EENC were twice as likely to report good practices as those with a negative attitude (AOR = 2.00, 95%CI: 1.17–3.41). Additionally, those who had received EENC training were 1.88 times more likely to report good EENC practices than those who had not (AOR = 1.88, 95%CI: 1.17–3.02). The univariable and binary logistic regression analyses for EENC practice are shown in Table 4.

**Table 3. Univariable and binary logistic regression analyses of the attitude towards EENC.**

| Predictors | Attitude | | COR (95%CI) | AOR (95%CI) |
|---|---|---|---|---|
| | Positive attitude | Negative attitude | | |
| Gender | | | | |
| Female | 352 (99.2) | 72 (92.3) | 1 | 1 |
| Male | 3 (0.8) | 6 (7.7) | 0.10 (0.03-0.42) ** | 0.11 (0.02-0.57)** |
| Age | | | | |
| <25 | 14 (3.9) | 4 (5.1) | 1 | 1 |
| 25-29 | 78 (22.0) | 28 (35.9) | 0.80 (0.24-2.62) | 0.41 (0.096-1.75) |
| 30-34 | 129 (36.3) | 34 (43.6) | 1.08 (0.34-3.51) | 0.48 (0.11-1.99) |
| ≥35 | 134 (37.7) | 12 (15.4) | 3.19 (0.91-11.23) | 1.43 (0.31-6.62) |
| Parental status | | | | |
| Without child | 125 (35.2) | 37 (47.4) | 1 | 1 |
| With child | 230 (64.8) | 41 (52.6) | 1.66 (1.01-2.72) * | 1.01 (0.47-2.15) |
| Perceived workload | | | | |
| Acceptable | 251 (70.7) | 39 (50) | 1 | 1 |
| Heavy | 104 (29.3) | 39 (50) | 0.41 (0.25-0.68) *** | 0.52 (0.28-0.94)* |
| Routine Workflow on EENC | | | | |
| Yes | 321 (90.4) | 63 (80.8) | 1 | 1 |
| No | 34 (9.6) | 15 (19.2) | 0.45 (0.23-0.87) * | 0.70 (0.30-1.67) |
| Position | | | | |
| None | 256 (72.1) | 71 (91.0) | 1 | 1 |
| Group leader | 86 (24.2) | 7 (9.0) | 3.41 (1.51-7.69) ** | 1.74 (0.65-4.69) |
| Nursing superior | 13 (3.7) | 0 (0) | -- | -- |
| Training on EENC | | | | |
| No | 84 (23.7) | 33 (42.3) | 1 | 1 |
| Yes | 271 (76.3) | 45 (57.7) | 2.37 (1.42-3.95) *** | 2.47 (1.31-4.67) ** |
| Years of working (years) | | | | |
| ≤7 | 107 (30.1) | 36 (46.2) | 1 | 1 |
| 8-14 | 160 (45.1) | 36 (46.2) | 1.50 (0.89-2.52) | 1.12 (0.43-2.91) |
| ≥15 | 88 (24.8) | 21 (7.7) | 4.94 (1.99-12.25) *** | 1.43 (0.28-7.35) |
| Knowledge of EENC | | | | |
| Poor knowledge | 102 (28.7) | 38 (48.7) | 1 | 1 |
| Good knowledge | 253 (71.3) | 40 (51.3) | 1.60 (1.29-1.98) *** | 2.23 (1.20-4.13)* |
| ICS | 3.45±0.71 | 2.63±0.67 | 4.03 (2.79-5.85) *** | 4.56 (2.98-6.99) *** |

*p<0.05, **p<0.01, ***p<0.001 COR: crude odds ratio; AOR: adjusted odds ratio; CI: confidence interval.

## 3.5. Pathways of implementation climate on EENC practice

Based on our hypothesis and the results from logistic regression analysis, the scores for implementation climate, training, knowledge, attitude, and practice of EENC were entered into Amos 24.0 for path analysis. The analysis revealed that implementation climate had statistically significant indirect effects on EENC practice through its influence on midwives' attitudes. EENC training had both direct and indirect effects on practice, impacting participants' knowledge and attitude. The results are illustrated in Fig 1. All paths in the final model had statistically significant coefficients, with robust fit

**Table 4. Univariable and binary logistic regression analyses of the practice of EENC.**

| Predictors | Practice | | COR (95%CI) | AOR (95%CI) |
|---|---|---|---|---|
| | Good n (%) | Poor n (%) | | |
| Age | | | | |
| <25 | 7 (2.3) | 11 (8.3) | 1 | 1 |
| 25-29 | 74 (24.7) | 32 (24.1) | 3.63 (1.29-10.22)* | 2.67 (0.88-8.13) |
| 30-34 | 106 (35.3) | 57 (42.9) | 2.92 (1.07-7.95) * | 1.85 (0.63-5.43) |
| ≥35 | 113 (37.7) | 33 (24.8) | 5.38 (1.93-14.98) ** | 3.03 (0.97-9.45) |
| Position | | | | |
| None | 213 (71.0) | 114 (85.7) | 1 | 1 |
| Group leader | 78 (26.0) | 15 (11.3) | 2.78 (1.53-5.06) *** | 2.24 (1.20-4.17)* |
| Nursing supervisor | 9 (3.0) | 4 (3.0) | 1.20 (0.36-4.00) | 1.71 (0.27-10.68) |
| Perceived workload | | | | |
| Acceptable | 209 (69.7) | 81 (60.9) | 1 | 1 |
| Heavy | 91 (30.3) | 52 (39.1) | 0.68 (0.44-1.04) | 0.74 (0.47-1.19) |
| Routine Workflow on EENC | | | | |
| Yes | 273 (91.0) | 111 (83.5) | 1 | 1 |
| No | 27 (9.0) | 22 (16.5) | 0.50 (0.27-0.91) * | 0.71 (0.36-1.42) |
| Training on EENC | | | | |
| No | 65 (21.7) | 52 (39.1) | 1 | 1 |
| Yes | 235 (78.3) | 81 (60.9) | 2.32 (1.49-3.62) *** | 1.88 (1.17-3.02)** |
| Years of working | | | | |
| ≤7 | 97 (32.3) | 46 (34.6) | 1 | 1 |
| 8-14 | 128 (42.7) | 68 (51.1) | 0.89 (0.57-1.41) | 0.65 (0.30-1.42) |
| ≥15 | 75 (25) | 19 (14.3) | 1.87 (1.01-3.46) * | 0.86 (0.28-2.65) |
| Attitude towards EENC | | | | |
| Negative attitude | 39 (13.0) | 39 (29.3) | 1 | 1 |
| Positive attitude | 261 (87.0) | 94 (70.7) | 1.09 (1.04-1.13) *** | 2.75 (1.76-4.31)*** |
| Knowledge of EENC | | | | |
| Poor knowledge | 72 (24.0) | 68 (51.1) | 1 | 1 |
| Good knowledge | 228 (76.0) | 65 (48.9) | 1.91 (1.56-2.34) *** | 2.00 (1.17-3.41) ** |

*p<0.05, **p<0.01, ***p<0.001; COR: crude odds ratio; AOR: adjusted odds ratio; CI: confidence interval.

indices: GFI = 0.997, AGFI = 0.977, NFI = 0.988, CFI = 0.995, TLI = 0.975, RMSEA = 0.039, and $X^2$/df = 1.658. The direct and indirect effects of the variables are summarized in Table 5.

## 4. Discussion

Our study highlights several critical factors that influence the practice of EENC among nurse-midwives, including implementation climate, training, knowledge, attitude, workload, and leadership roles. These factors not only independently affect EENC practice but also interact in complex ways to shape the quality of care.

### 4.1. The role of implementation climate and attitude

A key finding of our study is the indirect effect of implementation climate on EENC practice, mediated through its influence on midwives' attitudes. A positive organizational climate, characterized by supportive leadership and recognition, significantly improved

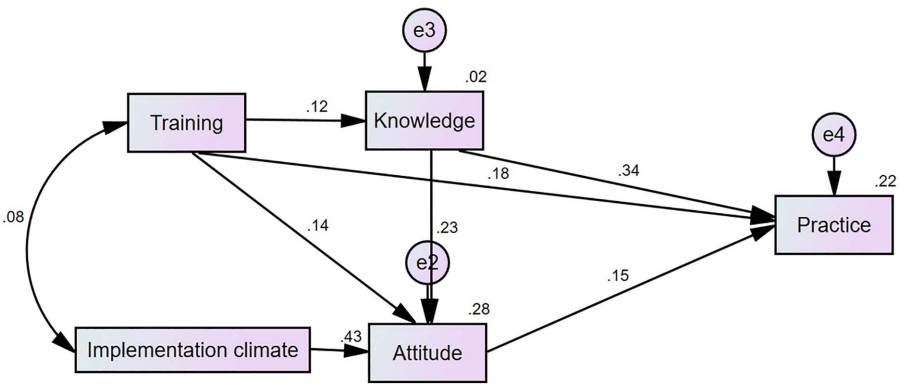

**Fig 1. Path analysis of the effect of implementation climate on EENC practice.**

**Table 5. Types of direct and indirect effects.**

|  | Direct effects | Indirect effects | Total effects |
|---|---|---|---|
| knowledge <--- training | 0.125 | -- | 0.125 |
| attitude <--- training | 0.140 | 0.028 | 0.169 |
| practice <--- training | 0.178 | 0.068 | 0.245 |
| attitude <--- implementation climate | 0.433 | -- | 0.433 |
| practice <--- implementation climate | -- | 0.064 | 0.064 |
| attitude <--- knowledge | 0.227 | -- | 0.227 |
| practice <--- knowledge | 0.342 | 0.034 | 0.375 |
| practice <--- attitude | 0.148 | -- | 0.148 |

attitudes, which in turn enhanced EENC practices. This aligns with existing literature suggesting that a supportive work environment fosters positive attitudes towards evidence-based practices [30], ultimately improving implementation outcomes. Therefore, improving the implementation climate—through recognition, rewards, and educational support—can enhance practice.

Path analysis confirmed that the implementation climate had a significant indirect effect on EENC practice through attitudes (RMSEA = 0.039). A one-point increase in the implementation climate score increased the odds of having a positive attitude by 4.56 times (AOR = 4.56, 95%CI: 2.98–6.99), which, in turn, predicted better practice (AOR = 2.00, 95%CI: 1.17–3.41). This supports Aarons' theoretical link between organizational climate to behavior change [31], and highlights climate as a critical factor in implementing evidence-based practices [32].

Notably, the rewards subscale of the implementation climate scored the lowest (3.02 ± 1.11), consistent with global findings that inadequate incentives hinder evidence-based practice [30]. To address this, integrating both financial and non-monetary rewards, such as elevating EENC champions, could improve motivation. This approach is supported by Schein's framework, which underscores the importance of leadership in reinforcing climate through tangible rewards [33].

Although the overall implementation climate scores in our study were higher than those in critical care settings [30], likely due to China's centralized training infrastructure through CMCHA hubs, the persistently low scores on the rewards dimension suggest that institutional incentives are needed to sustain guideline adherence.

## 4.2. Training as a key factor for knowledge and practice

Our results underscore the critical role of EENC training, which had both direct and indirect effects on practice. Nurse-midwives who received EENC training were more likely to demonstrate good knowledge and have a positive attitude,

which in turn contributed to better EENC practice. This highlights the essential role of training programs in equipping healthcare providers with the necessary skills and knowledge to effectively implement guidelines. This is consistent with another study from Afghanistan, which underscores ongoing training of healthcare workers to ensure consistent SSC practices [34]. Notably, knowledge and attitude were strong predictors of practice, education remains fundamental to successfully implementing EENC.

While 69.3% of participants demonstrated good EENC practice (Table 2), aligning with findings from Ethiopia (64%−72.7%) [35,36], critical gaps remain. Suboptimal adherence to practices such as umbilical cord care and routine eye care persists, consistent with previous surveys in China [37]. Lingering safety concerns may explain these deficits [38].

Our findings emphasize the need for updated guidelines and the dissemination of high-quality evidence to address knowledge gaps. Moreover, although most participants (88.5%) reported ensuring 90 minutes of skin-to-skin contact, the actual implementation in clinical settings requires further scrutiny to ensure that these practices are consistently carried out in practice.

### 4.3. Perceived workload and its impact on attitudes and practice

Our study identified that perceived heavier workloads were associated with more negative attitudes and poorer EENC practices. Specifically, nurse-midwives with heavier workloads were 48% less likely to hold a positive attitude towards EENC (AOR = 0.52, 95%CI: 0.28–0.94; Table 3). This finding is consistent with research suggesting that heavy workloads can lead to stress, burnout, and a lack of time for training and reflection, thereby undermining the consistent delivery of high-quality care practices [39]. Demanding workloads can specifically impair their capacity to follow through with evidence-based practices, particularly those that require attention to detail and time-sensitive actions [40].

Consequently, our findings underscore the importance of addressing workload management as a key strategy. Reducing workload and offering appropriate support are crucial not only for enhancing job satisfaction but also for enabling the reliable delivery of evidence-based care.

### 4.4. Leadership's role in shaping practice

Our study highlighted the significant role of group leaders in influencing EENC practice. Logistic regression analysis revealed that group leaders had 2.24-fold higher odds of demonstrating good EENC practice compared to non-leaders (AOR = 2.24, 95%CI: 1.20–4.17; Table 4). Group leaders, as middle managers, play a pivotal role in translating policy directives into frontline workflows, directly supervising 21.5% of the cohort (Table 1). Their influence is similar to findings in other fields, such as autism care, where middle managers' leadership indirectly enhanced the adoption of evidence-based practices [14]. By fostering trust and providing real-time feedback, group leaders help mitigate implementation resistance, a critical role often overlooked in innovation scale-up [41].

Empowering group leaders with leadership training and the necessary resources could enhance their ability to translate organizational priorities into daily practices. This in turn could amplify the implementation climate and foster a culture of quality care, where best practices are consistently followed. Ultimately, investing in middle management represents a key strategy for not only adopting but also maintaining EENC practices across healthcare settings.

### 4.5. Implications for practice and policy

The findings from this study suggest several practical implications for improving EENC implementation. First, efforts to improve implementation climate—especially through leadership support and appropriate rewards—should be prioritized in healthcare settings. Additionally, EENC training should be made more widely available and integrated into continuous

professional development programs. Lastly, reducing workload pressures and enhancing the support for mid-level leadership can help address barriers to optimal EENC implementation.

By focusing on these factors, healthcare systems can create an environment where EENC practices are more consistently applied, leading to better maternal and neonatal outcomes. This approach is particularly critical in low-resource settings, where systemic barriers are most prevalent.

### 4.6. Strengths and limitations

This study contributes to the understanding of EENC implementation in two key ways. First, the multi-center design—including 12 tertiary hospitals across China with diverse geographical and socio-economic backgrounds—enhances the generalizability of the findings to similar high-volume facilities in low- and middle-income countries. By including institutions from both economically developed eastern regions and underdeveloped western regions, we captured variations in resource availability and clinical workflows, ensuring the relevance of the results to heterogeneous healthcare settings. Second, the use of path analysis provides novel insights into the mechanisms linking implementation climate to practice, offering a deeper understanding of the direct and indirect effects that traditional regression models do not capture. This methodological innovation supports recent calls for advanced analytics in implementation science [42]. Lastly, our focus on middle leadership roles addresses a significant gap in maternal-neonatal research, offering actionable insights for policymakers aiming to scale EENC effectively.

However, there are some limitations to consider. Firstly, since the data were self-reported by nurse-midwives, there is a potential risk of social desirability bias, where participants may have provided more favorable responses than the actual situation. Future research should incorporate multiple data sources to mitigate this bias. Secondly, all participants were from tertiary hospitals, which limits the generalizability of the findings to secondary and primary healthcare settings. Additional studies are needed to assess EENC implementation in these settings and develop more comprehensive models considering different levels of healthcare contexts. Thirdly, the majority of participants in this study were female nurse-midwives, which aligns with the gender composition of the profession in the study setting but may limit insights into male nurses' perspectives on EENC. Future studies could intentionally include more male participants to explore potential gender differences in EENC-related attitudes and practices. Additionally, in the multivariate logistic regression analyses (Tables 3 and 4), certain categories—such as males with negative attitudes, participants aged <25 with negative attitudes, and nurse superiors with negative attitudes or poor practices—contained limited observations (n < 5). Such sparse data can potentially affect the stability and precision of the associations reported. Therefore, these results should be interpreted with caution. Further studies with larger sample sizes are needed to validate these associations with greater reliability. Finally, the cross-sectional design of this study precludes causal inferences, and longitudinal studies are required to examine how evolving implementation climates influence the long-term sustainability of practice.

### 5. Conclusion

This study identifies four strategic priorities for optimizing EENC implementation: fostering supportive environments, expanding competency-based training, rationalizing clinical workloads, and enhancing leadership engagement. The interconnected nature of these factors underscores the necessity for multidimensional interventions to align clinical operations with evidence-based protocols. Such integrated approaches demonstrate particular promise in resource-constrained settings, where systemic implementation barriers disproportionately affect care quality. By simultaneously addressing organizational and human capital challenges, healthcare systems can establish self-reinforcing cycles of improvement – advancing both adherence to EENC standards and health outcomes for maternal-neonatal dyads. Sustainable change thus depends on investing in both the workplace ecosystems and the caregiver, rather than isolated technical solutions.

## Innovations

(1)  First use of path analysis to disentangle climate's dual role (direct/indirect) in EENC adoption.

(2)  Middle leadership (Group Leaders) identified as unsung enablers—23% higher odds of good practice vs. non-leaders.

(3)  Actionable climate gaps: Poor reward systems undermine climate effectiveness, highlighting a fixable barrier.

## Supporting information

**S1 File.  Questionnaire.**
(DOCX)

**S2 File.  The knowledge, attitude and practice of EENC.**
(DOCX)

## Acknowledgments

We acknowledge Xin Du and Julian Heng (Remotely Consulting, Australia) who provided professional English-language editing of this article (Certificate No. 7Ic8Pi7Q).

## Author contributions

**Conceptualization:** Hongxiao He.

**Data curation:** Hongxiao He, Jiahe Li, Junying Li.

**Formal analysis:** Hongxiao He.

**Investigation:** Hongxiao He, Jiahe Li, Junying Li.

**Methodology:** Hongxiao He.

**Project administration:** Luxia Gong, Ruyan Pang, Xiu Zhu.

**Resources:** Jie Lu, Linlin Cao.

**Supervision:** Hong Lu, Jie Lu, Xiu Zhu.

**Writing – original draft:** Hongxiao He.

**Writing – review & editing:** Hong Lu, Xiu Zhu.

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
