## [Decision Letter · Decision Letter 0]

21 Jul 2025

PONE-D-25-27091The Role of Implementation Climate in Shaping Early Essential Newborn Care Practice: Insights from a Multi-Center Cross-Sectional Study in ChinaPLOS ONE

Dear Dr. Zhu,

Thank you for submitting your manuscript to PLOS ONE. After careful consideration, we feel that it has merit but does not fully meet PLOS ONE’s publication criteria as it currently stands. Therefore, we invite you to submit a revised version of the manuscript that addresses the points raised during the review process.

**Two review reports have been obtained. Please revise manuscript according to the reviewers' comments and my suggestions in the additional editor section comments. **

We look forward to receiving your revised manuscript.

Kind regards,

Muhammad Haroon Stanikzai

Academic Editor

PLOS ONE

**Journal Requirements:**

1. When submitting your revision, we need you to address these additional requirements. Please ensure that your manuscript meets PLOS ONE's style requirements, including those for file naming. The PLOS ONE style templates can be found at https://journals.plos.org/plosone/s/file?id=wjVg/PLOSOne_formatting_sample_main_body.pdf and https://journals.plos.org/plosone/s/file?id=ba62/PLOSOne_formatting_sample_title_authors_affiliations.pdf 2. Please provide additional details regarding participant consent. In the ethics statement in the Methods and online submission information, please ensure that you have specified (a) whether consent was informed and (b) what type you obtained (for instance, written or verbal, and if verbal, how it was documented and witnessed). If your study included minors, state whether you obtained consent from parents or guardians. If the need for consent was waived by the ethics committee, please include this information. If you are reporting a retrospective study of medical records or archived samples, please ensure that you have discussed whether all data were fully anonymized before you accessed them and/or whether the IRB or ethics committee waived the requirement for informed consent. If patients provided informed written consent to have data from their medical records used in research, please include this information. 3. We note that this data set consists of interview transcripts. Can you please confirm that all participants gave consent for interview transcript to be published? If they DID provide consent for these transcripts to be published, please also confirm that the transcripts do not contain any potentially identifying information (or let us know if the participants consented to having their personal details published and made publicly available). We consider the following details to be identifying information:- Names, nicknames, and initials- Age more specific than round numbers- GPS coordinates, physical addresses, IP addresses, email addresses- Information in small sample sizes (e.g. 40 students from X class in X year at X university)- Specific dates (e.g. visit dates, interview dates)- ID numbers Or, if the participants DID NOT provide consent for these transcripts to be published:- Provide a de-identified version of the data or excerpts of interview responses- Provide information regarding how these transcripts can be accessed by researchers who meet the criteria for access to confidential data, including:a) the grounds for restrictionb) the name of the ethics committee, Institutional Review Board, or third-party organization that is imposing sharing restrictions on the datac) a non-author, institutional point of contact that is able to field data access queries, in the interest of maintaining long-term data accessibility.d) Any relevant data set names, URLs, DOIs, etc. that an independent researcher would need in order to request your minimal data set. For further information on sharing data that contains sensitive participant information, please see: https://journals.plos.org/plosone/s/data-availability#loc-human-research-participant-data-and-other-sensitive-data If there are ethical, legal, or third-party restrictions upon your dataset, you must provide all of the following details (https://journals.plos.org/plosone/s/data-availability#loc-acceptable-data-access-restrictions):a) A complete description of the datasetb) The nature of the restrictions upon the data (ethical, legal, or owned by a third party) and the reasoning behind themc) The full name of the body imposing the restrictions upon your dataset (ethics committee, institution, data access committee, etc)d) If the data are owned by a third party, confirmation of whether the authors received any special privileges in accessing the data that other researchers would not havee) Direct, non-author contact information (preferably email) for the body imposing the restrictions upon the data, to which data access requests can be sent 4. Please include captions for your Supporting Information files at the end of your manuscript, and update any in-text citations to match accordingly. Please see our Supporting Information guidelines for more information: http://journals.plos.org/plosone/s/supporting-information. 5.If the reviewer comments include a recommendation to cite specific previously published works, please review and evaluate these publications to determine whether they are relevant and should be cited. There is no requirement to cite these works unless the editor has indicated otherwise. 

**Additional Editor Comments:**

- Abstract: It is better to write results section paragraph-vise rather to present in a bullet manner.

- The authors should proof read the entire article for language correction and grammar.

- Please look at published papers on PLOS ONE for in-text citation. The authors should a single space between the in-text citation and text. In most cases, they are now attached.

- Tables 3 & 4: Radio should be ratio.

- Line 104: Settings should be with small letter.

- Discussion: Please make only the first letter of subheading capital.

- Line 337-339: Add the following citations; https://journals.plos.org/plosone/article?id=10.1371/journal.pone.0324758

- References: Please review the reference and set them as per journal styles. References 1 and 4 are incomplete. Please provide availability link and access date.

- I noticed color changes in the text. Please use black color for all text.

Best of Luck.

Reviewers' comments:

Reviewer's Responses to Questions

**Comments to the Author**

1. Is the manuscript technically sound, and do the data support the conclusions?

Reviewer #1: Yes

Reviewer #2: Yes

2. Has the statistical analysis been performed appropriately and rigorously? 

Reviewer #1: Yes

Reviewer #2: Yes

3. Have the authors made all data underlying the findings in their manuscript fully available?

Reviewer #1: Yes

Reviewer #2: Yes

4. Is the manuscript presented in an intelligible fashion and written in standard English?

Reviewer #1: Yes

Reviewer #2: Yes

5. Review Comments to the Author

**Reviewer #1:**  thank you for sharing a very important study on EENC.

Abstract: it is better to remove bullet numbering in your results section: (1)..., (2)...., (3).... please write your results in continues prose.

methodology

your tools for measurement of attitude was developed by authors? if yes please give more information about its validity.

Results

Table 3, 4: COR and AOR; clarify abbreviations at first mention, and please write ratio instead of radio.

Thank you

**Reviewer #2:**  Dear Editors and Authors,

Thank you for giving me the opportunity to review the manuscript, under the title “The Role of Implementation Climate in Shaping Early Essential Newborn Care (EENC) Practice: Insights from a Multi-Center Cross-Sectional Study in China”.

The authors chose a very important topic on newborn care in China. This study was conducted between Dec 2022 and April 2023 among nurse-midwives from 12 tertiary-level maternity facilities across China. The authors developed and validated a structured questionnaire, using the WHO guidelines. The study examined EENC-related knowledge, attitudes, practices, and perceived implementation climate, using path analysis and logistic regression to assess direct and indirect associations. The authors reported prevalence of good EENC practice, mean score of implementation climate, and found that good knowledge, positive attitudes, training, leadership role, and perceived workload to be predictors of good EENC practice (with heavier workloads having a negative effect on positive attitude, which impacted good EENC practice, and with good knowledge, in-service training, positive attitude, and leadership role having a positive effect on good EENC practice).

Better implementation climate had positive effects on attitude, and implementation climate positively affected EENC practice indirectly through attitudes.

The authors concluded that both individual factors and organizational climate are important influential factors of EENC practices. To improve EENC implementation, healthcare systems should focus on enhancing the implementation climate through leadership support, training staff, appropriate reward systems, and addressing workload challenges.

My overall assessment of the manuscript is that it has been written well and can be considered for publication. There are some minor issues that the authors need to address.

1. In table3 and table 4, there is an inconsistency in using “bivariable” and “univariable”. The authors can use either one; but this should be consistent throughout the manuscript.

2. For figure1, some words start with capital letters; while others start with small letters (e.g., Training, knowledge). They need to be consistent – either start with small or capital letters.

3. Oversampling of female nurse-midwives can be a potential limitation in this study. The authors may opt to discuss it in the limitation of this study, because male nurses may have different attitudes and practices with respect to EENC, although I understand that midwives are female and this sample can present a valid representation of those health workers, who are mainly involved in newborn care.

6. PLOS authors have the option to publish the peer review history of their article (what does this mean? ). If published, this will include your full peer review and any attached files.

**Do you want your identity to be public for this peer review?** For information about this choice, including consent withdrawal, please see our Privacy Policy .

Reviewer #1: No

Reviewer #2: No

---

## [Author Response · Author response to Decision Letter 1]

3 Sep 2025

Dear Editors and Reviewers:

Thank you for your letter and for the time you have taken to review our manuscript. We greatly appreciate your valuable feedback, which has been essential in improving our paper. We have carefully studied all comments and have incorporated revision accordingly. The main corrections and our point-by point response are provided below.

Reviewer 1

Abstract: it is better to remove bullet numbering in your results section: (1)..., (2)...., (3).... please write your results in continues prose.

Response: We thank you for this suggestion. The results section of the abstract has been revised into continuous prose by removing the bullet points, which improves the flow and readability.

Methodology: your tools for measurement of attitude was developed by authors? if yes please give more information about its validity.

Response: Thank you for raising this important point. Yes, the attitude measurement tool was developed by the authors. Its validity was supported through a combination of qualitative insight, expert review, and statistical evaluation.

The tool was designed based on qualitative fieldwork, expert review, and psychometric testing. Specifically, we included items 4 and 5 to capture locally relevant attitudes toward two controversial EENC practices—non-routine cord disinfection and non-traditional suctioning—which were frequently raised by midwives in our context. Although Items 4, 5, and 10 had modest factor loading, they were retained due to their contextual and theoretical relevance. Item 10, in particular, reflects institutional motivation and perceived competitive pressure, a dimension not covered by other items.

Except for these three items, all other items exhibited standardized regression weights ranging from 0.596 to 0.905, exceeding the conventional threshold of 0.5, thus indicating satisfactory convergent validity for the latent construct of attitude.

Below are the standardized regression weights for all items:

Estimate

Item 1<---Attitude 0.891

Item 2<---Attitude 0.898

Item 3<---Attitude 0.834

Item 4<---Attitude 0.253

Item 5<---Attitude 0.414

Item 6<---Attitude 0.905

Item 7<---Attitude 0.629

Item 8<---Attitude 0.596

Item 9<---Attitude 0.687

Item 10<---Attitude 0.322

We hope this clarification adequately addresses your question regarding the validity of the attitude scale.The corresponding modifications have been incorporated into the revised manuscript on line 164-173.

Results: Table 3, 4: COR and AOR; clarify abbreviations at first mention, and please write ratio instead of radio.

Response:Thank you for these valuable corrections. We have clarified the abbreviations "COR" (Crude Odds Ratio) and "AOR" (Adjusted Odds Ratio) at their first mention in the text (Line 282 and Line 200, respectively). Additionally, the typographical error of "radio" has been corrected to "ratio".

Reviewer 2

In table3 and table 4, there is an inconsistency in using “bivariable” and“univariable”. The authors can use either one; but this should be consistent throughout the manuscript.

Response:We sincerely thank the reviewer for this careful observation and valuable comment. We apologize for this inconsistency in our terminology.We have now revised the manuscript to consistently use the term "univariable" throughout, including in Tables 3 and 4 and in the corresponding descriptions in the main text.

2. For figure1, some words start with capital letters; while others start with small letters (e.g., Training, knowledge). They need to be consistent – either start with small or capital letters.

Response:Thank you for this helpful feedback. We have revised Figure 1 to ensure consistent capitalization (all words now begin with a capital letter) and have re-uploaded the updated figure.

3. Oversampling of female nurse-midwives can be a potential limitation in this study. The authors may opt to discuss it in the limitation of this study, because male nurses may have different attitudes and practices with respect to EENC, although I understand that midwives are female and this sample can present a valid representation of those health workers, who are mainly involved in newborn care.

Response:We thank you for this insightful comment. We agree that the predominance of female nurse-midwives in our sample is an important point to address. Although this reflects the real-world gender distribution of midwifery professionals in the study context—where midwives are predominantly female—we acknowledge that including more male nurses could provide additional perspectives on EENC attitudes and practices. We have added this point to the limitations section on line 425-429 to offer a more comprehensive discussion of the generalizability of our findings.

Additional Editor Comments

Abstract: It is better to write results section paragraph-vise rather to present in a bullet manner.

Response:Thank you for the suggestion. We have revised the results section of the abstract into a paragraph format as recommended.

The authors should proof read the entire article for language correction and grammar.

Response:Thank you for this comment.Our team has carefully re-examined the manuscript to address grammatical issues and refine the expression. We would be grateful if the reviewer could point out any specific passages that still require attention.

Please look at published papers on PLOS ONE for in-text citation. The authors should a single space between the in-text citation and text. In most cases, they are now attached.

Response:Thank you for this meticulous feedback. We have carefully reviewed the entire manuscript and ensured that a single space is now consistently applied between all in-text citations and the preceding text.

Tables 3 & 4: Radio should be ratio.

Response: We thank the reviewer for pointing out this typographical error. We have corrected "Radio" to "Ratio" in both Table 3 and Table 4. We apologize for any confusion this may have caused.

Line 104: Settings should be with small letter.

Response:Thank you for pointing this out. The word on line 104 has been corrected to "settings" as suggested. We have also taken care to ensure consistent capitalization throughout the manuscript.

Discussion: Please make only the first letter of subheading capital.

Response: Thank you for pointing this out. We have revised the subheading in the Discussion section to have only the first letter capitalized.

Line 337-339: Add the following citations https://journals.plos.org/plosone/article?id=10.1371/journal.pone.0324758

Response: We agree with your suggestion. The recommended citation has been added to the discussion section (lines 343–345), as it provides valuable external validation for our argument on the necessity of ongoing training.

The added sentence reads: “This is consistent with another study from Afghanistan, which underscores ongoing training of healthcare workers to ensure consistent SSC practices (34). ”

References: Please review the reference and set them as per journal styles. References 1 and 4 are incomplete. Please provide availability link and access date.

Response: Thank you for this suggestion. We have carefully reviewed and reformatted all references according to the journal’s style guidelines. In particular, references 1 and 4 have been completed with the required access links and retrieval dates.

I noticed color changes in the text. Please use black color for all text.

Response: Thank you for pointing this out. We have revised the manuscript to use black text throughout and ensured all text now consistently appears in black.

---

## [Editor Report · Decision Letter 1]

8 Sep 2025

PONE-D-25-27091R1The Role of Implementation Climate in Shaping Early Essential Newborn Care Practice: Insights from a Multi-Center Cross-Sectional Study in ChinaPLOS ONE

Dear Dr. Zhu,

Thank you for submitting your manuscript to PLOS ONE. After careful consideration, we feel that it has merit but does not fully meet PLOS ONE’s publication criteria as it currently stands. Therefore, we invite you to submit a revised version of the manuscript that addresses the points raised during the review process.

**Thank you for positively responding to esteemed reviewers' comments and suggestions. I request some minor corrections before the manuscript can be accepted for publication. You can find my suggestions at additional editor comments section.**==============================

We look forward to receiving your revised manuscript.

Kind regards,

Muhammad Haroon Stanikzai

Academic Editor

PLOS ONE

Journal Requirements:

Additional Editor Comments:

- For PLOS ONE publication, we use [] for in-text citation. Please revise across the documents.

- Line 301: Please put figure legend below the figure. Please upload Figure 1 as Per PLOS One style using https://ngplosjournals.pagemajik.ai/artanalysis. Delete figure 1 from document and upload as figure file after following journal guidelines. Only leave figure legend in the manuscript document.

- Tables 3 and 4. There is no single space between frequency and percentage brackets in some rows. Be consistent.

- In tables 3 and 4, some cells have value less than 5, which increases the chances of uncertainty. It is better to correct or state as a limitation of the study.

- The term ‘early essential newborn care (EENC)’ was used in the manuscript several times. Once a term is abbreviated for the first time, it is not necessary to use the abbreviated form with its full form repeatedly. I can see the term is abbreviated in lines 226, 233, 307, and 404. Only the abbreviated term EENC can be used subsequently across the document.

---

## [Author Response · Author response to Decision Letter 2]

27 Sep 2025

Dear Editors and Reviewers:

Thank you for your letter and for the time you have taken to review our manuscript. We greatly appreciate your valuable feedback, which has been essential in improving our paper. We have carefully studied all comments and have incorporated revision accordingly. The main corrections and our point-by point response are provided below.

Journal Requirements:

Response: Yes, we received a recommendation to cite a specific published study. We have reviewed the suggested paper, which examines the prevalence and factors associated with mother-newborn skin-to-skin contact in Afghanistan, and found it highly relevant to our work--particularly in supporting our findings related to training. Accordingly, we have cited this reference in the revised manuscript (lines 371-373).

Response: We thank the editor for highlighting this important step. We have thoroughly reviewed all references in our manuscript and confirmed that none have been retracted. We have reviewed the reference list to ensure completeness and accuracy.

Additional Editor Comments:

1. For PLOS ONE publication, we use [] for in-text citation. Please revise across the documents.

Response: Thank you for pointing this out. We have revised all in-text citations throughout the manuscript to use square brackets in accordance with PLOS ONE style requirements.

2. Line 301: Please put figure legend below the figure. Please upload Figure 1 as Per PLOS One style using https://ngplosjournals.pagemajik.ai/artanalysis. Delete figure 1 from document and upload as figure file after following journal guidelines. Only leave figure legend in the manuscript document.

Response: Thank you for this guidance. We have removed Figure 1 from the manuscript document and uploaded it as a separate figure file following the journal’s guidelines.

3.Tables 3 and 4. There is no single space between frequency and percentage brackets in some rows. Be consistent.

Response: Thank you for this meticulous feedback. We have carefully reformatted Tables 3 and 4 to ensure consistent single spacing between frequency values and percentage brackets throughout.

4. In tables 3 and 4, some cells have value less than 5, which increases the chances of uncertainty. It is better to correct or state as a limitation of the study.

Response: We appreciate this suggestion. We have stated this as a limitation of the study in the revised manuscript (line 456-463).

5. The term ‘early essential newborn care (EENC)’ was used in the manuscript several times. Once a term is abbreviated for the first time, it is not necessary to use the abbreviated form with its full form repeatedly. I can see the term is abbreviated in lines 226, 233, 307, and 404. Only the abbreviated term EENC can be used subsequently across the document.

Response: Thank you for pointing this out. We have revised the manuscript to use the abbreviated form “EENC” after its first full introduction. The full term with abbreviation now appears only once, and the abbreviated form is used consistently thereafter.

---

## [Editor Report · Decision Letter 2]

2 Oct 2025

The Role of Implementation Climate in Shaping Early Essential Newborn Care Practice: Insights from a Multi-Center Cross-Sectional Study in China

PONE-D-25-27091R2

Dear Dr. Zhu,

We’re pleased to inform you that your manuscript has been judged scientifically suitable for publication and will be formally accepted for publication once it meets all outstanding technical requirements.

Kind regards,

Muhammad Haroon Stanikzai

Academic Editor

PLOS ONE

Additional Editor Comments (optional):

Congratulations on this wonderful work. 
---

## [Editor Report · Acceptance letter]

PONE-D-25-27091R2

PLOS ONE

Dear Dr. Zhu,

I'm pleased to inform you that your manuscript has been deemed suitable for publication in PLOS ONE. Congratulations! Your manuscript is now being handed over to our production team.

Kind regards,

on behalf of

Dr. Muhammad Haroon Stanikzai

Academic Editor

PLOS ONE